# Coverage Strategy for Small-Cell UAV-Based Networks in IoT Environment

**DOI:** 10.3390/s23218771

**Published:** 2023-10-27

**Authors:** Mohamed Ould-Elhassen Aoueileyine, Ramzi Allani, Ridha Bouallegue, Anis Yazidi

**Affiliations:** 1Innov’COM Laboratory, Higher School of Communication of Tunis, Ariana 2083, Tunisia; ramzi.allani@supcom.tn; 2Department of Computer Science, OsloMet—Oslo Metropolitan University, 0176 Oslo, Norway

**Keywords:** UAV, small cells, Nash equilibrium, encounter rate, beaconing period, coverage, IoT

## Abstract

In wireless communication, small cells are low-powered cellular base stations that can be used to enhance the coverage and capacity of wireless networks in areas where traditional cell towers may not be practical or cost-effective. Unmanned aerial vehicles (UAVs) can be used to quickly deploy and position small cells in areas that are difficult to access or where traditional infrastructure is not feasible. UAVs are deployed by telecommunication service providers to provide aerial network access in remote rural areas, disaster-affected areas, or massive-attendance events. In this paper, we focus on the scheduling of beaconing periods as an efficient means of energy consumption optimization. The conducted study provides a sub-modular game perspective of the problem and investigates its structural properties. We also provide a learning algorithm that ensures convergence of the considered UAV network to a Nash equilibrium operating point. Finally, we conduct extensive numerical investigations to assist our claims about the energy and data rate efficiency of the strategic beaconing policy (at Nash equilibrium).

## 1. Introduction

In recent years, unmanned aerial vehicles (UAVs), commonly known as drones, have garnered significant attention due to their versatility, agility, and cost-effective deployment [1]. Equipped with advanced navigation systems and smart sensors, UAVs are currently employed in a range of applications, including surveillance, search-and-rescue missions, and on-demand communication services. As UAV technology matures and regulations evolve, the global UAV market is poised for substantial growth [2,3,4].

UAVs offer a promising solution for supporting cellular communication networks in scenarios where terrestrial infrastructure is compromised. They excel at providing wireless communication services in hard-to-reach rural areas, at large-scale events such as festivals and sports gatherings, and during emergency situations where traditional base-station installation is prohibitively expensive. Leveraging UAVs as flying base stations offers several advantages, including exceptional maneuverability, adaptable deployment, on-demand telecommunication efficiency, and mobility enhancements [5].

The inherent mobility and aerial positioning of UAVs contribute to reliable communication channels, with the UAV–ground link frequently establishing Line-of-Sight (LoS) connections [6]. Furthermore, UAVs play a pivotal role in the Internet of Things (IoT), a transformative technology that has transitioned from concept to reality [7,8]. The IoT enables seamless data exchange and interoperability among devices within the internet infrastructure, providing pervasive connectivity, reducing transmission costs, and extending the reach of low-power communication. UAVs, with their flexible deployment options and mission adaptability, play a crucial role in realizing the IoT vision and offer a wide array of solutions and services to IoT partners [9].

Despite the numerous advantages of deploying drones as flying base stations, several economic and technical challenges must be addressed. These challenges encompass technical and physical hurdles, pricing strategies, and the availability of management features. To tackle these pressing issues, the scheduling of UAV availability or beaconing becomes a crucial but relatively unexplored aspect. Consequently, there is a growing need for comprehensive modeling and performance analysis of UAV setups to address these challenges effectively [10].

A novel approach that combines long short-term memory (LSTM) and deep reinforcement learning (DRL) techniques to tackle the multi-objective optimization problem is used. By leveraging LSTM-based DRL, the authors aim to achieve efficient resource allocation, maximize energy transfer efficiency, minimize latency, and enhance the overall performance of UAV-enabled wirelessly powered IoT networks. The proposed approach contributes to the field by providing a promising solution that considers multiple objectives and optimizes resource allocation dynamically, taking into account the changing network conditions [11,12,13]. This paper explores the potential of integrating UAVs with IoT networks and examines the unique characteristics and capabilities that UAVs bring to the IoT ecosystem. It highlights the opportunities presented by UAV-enabled IoT networks in various application domains, such as disaster management, precision agriculture, environmental monitoring, and smart cities. Additionally, this paper discusses the challenges and issues that need to be addressed for the successful deployment and operation of UAV-enabled IoT networks, including energy efficiency, connectivity, security, privacy, and regulatory considerations. The contribution of this paper lies in providing valuable insights into the architecture, opportunities, and challenges associated with UAV-enabled IoT networks, paving the way for future research and development in this emerging field [14,15,16]. This paper focuses on a practical energy consumption model for UAVs and aims to optimize the energy allocation and transmission policies to maximize the overall network performance. By considering the energy limitations of a UAV, this paper aims to find efficient solutions that ensure reliable wireless power transfer to IoT devices while maximizing the UAV’s flight time and minimizing the energy consumption. The proposed optimization framework and energy consumption model contribute to the field by providing insights into the design and operation of wirelessly powered IoT networks with energy-limited UAVs, facilitating the deployment of efficient and sustainable IoT applications [17,18,19]. This paper focuses on optimizing the data collection process by proposing a cluster-based approach that leverages UAVs for efficient communication and data gathering. The goal is to minimize energy consumption, maximize network lifetime, and improve data collection efficiency. This paper presents an optimization study that explores different parameters and factors affecting the performance of UAV-assisted cluster-based Wireless Sensors Network (WSNs). By identifying the optimal configuration and parameters, the proposed approach contributes to the field by providing insights into the efficient data collection in WSNs with UAV assistance in 3D environments, facilitating the deployment of reliable and energy-efficient sensor networks for various applications, such as environmental monitoring and surveillance [20]. This paper explores two different approaches to optimizing the AoI: off-policy and on-policy. Off-policy optimization refers to the optimization of the AoI based on historical data and past experiences, while on-policy optimization involves making real-time decisions to minimize the AoI. This paper compares these two approaches and investigates their effectiveness in UAV-RIS-assisted IoT networks. By examining the trade-offs between off-policy and on-policy optimization techniques, this paper contributes to the field by providing insights into the optimal strategies for minimizing the AoI in UAV-RIS-assisted IoT networks. The findings of this study can assist in the design and implementation of efficient and real-time data transmission protocols in IoT networks, improving the freshness and reliability of information [21]. In this paper, we delve into the intricate challenge of optimizing pricing and availability for UAVs, taking into account their limited battery capacity. Our primary focus is on the scheduling of availability periods, a pivotal factor that holds the potential to enhance energy efficiency, extend UAV operational lifetime, and ensure a satisfactory quality of service. Specifically, our research centers on addressing the joint pricing–availability problem within UAV-based networks. We introduce a novel approach that simultaneously tackles the issues of UAV pricing and energy efficiency, as documented in [22].

To grapple with these complex challenges, we investigate the intricate interplay between availability and pricing among service providers within the UAV ecosystem, treating it as a non-cooperative game. Our study adopts a duopoly framework, where a finite network of mobile unmanned aerial vehicles is deployed based on a homogeneous Poisson point process (PPP) to serve ground-based IoT devices. These UAVs navigate according to a random waypoint (RWP) model, as described in [23]. As an initial step, we derive expressions for the coverage probability and service probability of each UAV in the current scenario. To optimize the system’s overall performance in terms of pricing policies and energy efficiency, our approach incorporates a Nash equilibrium analysis. Additionally, we propose the use of learning automata as a strategic learning strategy to establish a joint price–availability equilibrium. Finally, we present extensive numerical simulations that underscore the significance of considering price and availability as interconnected decision parameters, offering valuable insights and heuristics for setting them optimally.

The remainder of this article is structured as follows: In Section 2, we provide an overview of the proposed UAV duopoly system model and our strategic approach to pricing and availability. Section 3 delves into the analysis of the availability game, addressing the existence and uniqueness of the Nash equilibrium solution and defining the game’s sub-modularity characteristics. Section 4 covers the numerical implementation and the pricing game, accompanied by a mathematical and analytical discussion of the property of super-modularity. Section 5 explores a joint availability–pricing approach with numerical learning implementations, highlighting the impact of various parameters on the learning process. Finally, in Section 6, we conclude our paper and suggest avenues for future research, building on the foundations laid out in this study.

## 2. Related Works

A significant existing body of literature investigates interesting features of UAV technological performance and reducing cost. For instance, Arabi, S. et al. [4] discuss the problem of data gathering and energy transfer in a UAV-assisted flying access network for IoT applications. The authors propose a joint optimization scheme that aims to minimize the energy consumption of the network while ensuring that the data gathering and energy transfer requirements are met. The proposed scheme involves the deployment of UAVs that act as flying base stations for IoT devices. These UAVs use energy transfer techniques, such as wireless power transfer, to recharge the IoT devices and prolong their operational lifetime. The authors consider two scenarios: one where the UAVs are stationary and one where they are mobile. To optimize the energy consumption of the network, the authors propose a two-stage optimization algorithm that jointly optimizes the UAVs’ trajectories and the energy transfer parameters. The algorithm takes into account the energy consumption of the UAVs, the energy consumption of the IoT devices, and the energy transfer efficiency. The authors evaluate the proposed scheme using simulations and show that it outperforms other existing schemes in terms of energy consumption and data gathering efficiency. They also discuss the limitations of the proposed scheme and identify future research directions. Aya Moheddine et al. [5] propose a novel solution for wireless connectivity in the Internet of Flying Things (IoFT). The authors propose using unmanned aerial vehicles (UAVs) as flying gateways for Long Range Wide Area Network (LoRaWAN) technology to overcome the challenges of wireless connectivity in the IoFT. The proposed solution involves equipping a UAV with an LoRaWAN gateway and using it as a mobile base station to provide wireless connectivity to IoT devices. The authors describe the technical details of the proposed solution, including the hardware and software architectures, and discuss the challenges of deploying a flying gateway in the IoFT. The authors conducted a proof-of-concept experiment to demonstrate the feasibility of their solution. The experiment involved deploying a UAV with an LoRaWAN gateway in a rural area and connecting it to a network of IoT devices. The authors measured the signal strength and data transmission rate and compared the results to a traditional stationary gateway. The results showed that the UAV-based gateway achieved better signal strength and higher data transmission rates than the stationary gateway. To investigate connectivity issues in depth, the authors of [24] address the challenge of improving the connectivity and coverage of flying networks, which consist of a set of unmanned aerial vehicles (UAVs) that are deployed to provide wireless communication services over a geographic area. The authors propose a new relay-positioning algorithm that takes into account both the energy consumption and the performance of the network, in order to maximize the network’s overall performance while minimizing the energy consumption of the UAVs. The proposed algorithm considers the mobility of the UAVs, as well as the quality of the wireless links among them, in order to determine the optimal placement of relays. The authors also consider the impact of interference and the energy consumption of the UAVs on the performance of the network. The paper presents simulation results that demonstrate the effectiveness of the proposed algorithm in improving the performance of flying networks compared with other existing relay-positioning algorithms. The authors also discuss the potential applications of their work in areas such as disaster response, surveillance, and transportation. The problem of power-splitting relaying protocol for simultaneous wireless information and power transfer (SWIPT) in a downlink non-orthogonal multiple-access (NOMA) Internet of Things (IoT) network with multiple unmanned aerial vehicles (UAVs) is described and detailed in [25]. The paper addresses the challenge of improving the energy efficiency and communication reliability of IoT networks by using UAVs as relays for power and data transmission. The authors propose a novel power-splitting relaying protocol that enables efficient energy harvesting by IoT devices while maintaining a reliable communication link with the UAV relays. The proposed protocol uses NOMA technology to allow multiple IoT devices to share the same frequency and time resources for communication, while the UAVs serve as relays for both power and data transmission. The power-splitting relaying protocol divides the signal received by each UAV into two parts: one for energy harvesting and the other for forwarding data to the IoT devices. The authors also consider the impact of interference and path loss on the performance of the network, and propose a joint power allocation and user grouping algorithm to optimize the performance of the system. The paper presents simulation results that demonstrate the effectiveness of the proposed power-splitting relaying protocol in improving the energy efficiency and communication reliability of the NOMA-IoT network with multiple UAVs compared with other existing protocols. The authors also discuss the potential applications of their work in areas such as smart cities, smart agriculture, and environmental monitoring. Huu Q. Tran et al. [26] propose a strategic and cost-effective solution for the deployment of unmanned aerial vehicle (UAV)-based flying access networks. The paper addresses the challenge of providing reliable and cost-effective wireless connectivity in areas with limited or no traditional infrastructure, such as rural or disaster-stricken regions. The authors propose a solution that involves deploying a fleet of UAVs equipped with wireless access points to provide connectivity to ground users. The proposed solution uses game theory to model the interactions between the UAVs and the ground users and optimizes the deployment strategy to achieve a balance between the availability and cost of the network. Specifically, the authors use s-modular game analysis, a variant of game theory, to model the strategic interactions between the UAVs and ground users in terms of availability and cost. Zhengyu Zhu et al. [27] propose a fuzzy logic-based solution for efficient UAV positioning in an Internet of Things (IoT) environment for data collection. The paper addresses the challenge of efficiently collecting data from IoT devices located in a wide area using a UAV as a data collector. The authors propose a fuzzy logic-based approach to determine the optimal position of the UAV for data collection, taking into account factors such as the number of IoT devices, the distance between the UAV and the devices, and the data transfer rate. The proposed solution involves designing a fuzzy logic controller that takes inputs such as the number of IoT devices, the distance between the UAV and the devices, and the data transfer rate and outputs the optimal position of the UAV for data collection. The authors also use a genetic algorithm to optimize the fuzzy logic controller parameters. As a solution to jointly optimizing the power allocation and 3D trajectory of a UAV in a wirelessly powered communication network with obstacles, the authors of [28] consider a scenario where a UAV is used as a mobile base station to provide wireless power transfer and communication services to ground users. The UAV is equipped with a wireless power transmitter and a communication antenna, and it moves along a 3D trajectory to serve users in the coverage area. However, the presence of obstacles in the environment can cause signal attenuation and interference, which affect the UAV’s performance. To address these challenges, the authors propose a joint power-and-3D-trajectory optimization scheme that takes into account the impact of obstacles on the UAV’s performance. The proposed solution involves formulating an optimization problem that aims to maximize the minimum achievable data rate of all users, subject to constraints on the UAV’s maximum transmit power, energy harvesting efficiency, and collision avoidance. The authors use a successive convex approximation (SCA) algorithm to solve the optimization problem and obtain the optimal power allocation and 3D trajectory of the UAV. The SCA algorithm involves iteratively solving a series of convex subproblems that approximate the original non-convex problem. The authors also propose a heuristic algorithm for collision avoidance that ensures that the UAV does not collide with obstacles while moving along the optimized trajectory. In Table 1, we present an inclusive overview of paramount research works across distinct dimensions in the realm of coverage strategy for small-cell UAV-based networks. The table is segmented into four distinct sections, each contributing to a holistic understanding of the field’s progression and multi-disciplinary nature:AI category: This section compiles seminal papers delving into the application of Artificial Intelligence (AI) within the context of our research area. By cataloging authors, publication dates, and central AI themes, this segment elucidates how AI methodologies have been harnessed to optimize coverage strategies for UAV-based networks in IoT environments.Big Data analytics: In this section, we assemble key contributions that explore the role of Big Data analytics within our research domain. By highlighting influential authors and pivotal concepts in the realm of data analysis, this section underscores the significance of data-driven insights in shaping effective coverage strategies.Key enabling technologies: This segment encapsulates pivotal works focusing on the foundational technologies that empower coverage strategies in small-cell UAV-based networks. With emphasis on authors and essential technological facets, this section portrays the intricate tapestry of technologies contributing to the operational efficacy of our research scope.Application domains: Lastly, we curate a selection of papers that elucidate the diverse application domains where our research finds relevance. This section unveils authors, publication chronology, and specific application contexts, illustrating how the proposed coverage strategies align with and enhance various IoT environments.

**Table 1 sensors-23-08771-t001:** Qualitative comparison of related works.

References	Year	AI Category	Big Data Analytics	Key Enabling Technologies	Application Domains
[29]	2020	Yes; fuzzy logic-based approach for efficient UAV positioning.	Yes; processing, analysis, and interpretation of the collected data.	Yes; flying base stations (FBSs) in an IoT environment.	Yes; optimization of the placement of drones as base stations to ensure full coverage of sensors and actuators.
[30]	2023	Yes; optimization in UAV-enabled wirelessly powered communication networks.	No.	Yes; UAVs, wirelessly powered communication networks, and optimization techniques.	Yes; disaster management, surveillance, or remote sensing.
[31]	2022	No.	No.	Yes; unmanned aerial vehicles (UAVs) as edge servers in mobile edge networks.	Yes; Quality-of-Service (QoS) in cluster-based UAV-assisted edge networks, specifically targeting the Internet of Things (IoT) domain.
[32]	2023	Yes; deep reinforcement learning (DRL) algorithm, deep deterministic policy gradient (DDPG).	No.	Yes; UAVs, wirelessly powered communication terminals, solar-powered UAVs.	Yes; Quality of Service (QoS).
[33]	2021	No.	No.	Yes; mobile edge computing, wireless power transmission, and UAVs for IoT networks.	Yes; UAV-based wireless power transmission and collaborative MEC to optimize the performance of IoT devices.
[34]	2022	No.	No.	Yes; non-orthogonal multiple access (NOMA), unmanned aerial vehicles (UAVs), and mobile edge computing (MEC) in the context of secure communication.	Yes; security aspects of NOMA-based communication in the presence of a flying eavesdropper.
[35]	2021	No.	No.	Yes; UAVs, multi-access edge computing	Yes; wireless networks, UAV-BSs.
[36]	2022	Yes; industrial knowledge graph-based relation mining, federated learning-based service prediction, and globally optimized resource reservation.	Yes; processing, analysis, and interpretation of the collected data	Yes; federated learning in Industrial IoT	Yes; Industrial IoT.
[37]	2021	Yes; distributed aggregation-based dispersed federated learning.	No.	Yes; federated learning for edge networks	Yes; strict latency Internet of Things (IoT) applications.

By structuring the analysis of related work in these meticulously crafted sections, we aim to offer readers a nuanced perspective on the multiple facets that collectively constitute the landscape of coverage strategies for small-cell UAV-based networks. This comprehensive tableau serves not only to contextualize our research but also to map the progression of ideas and innovations that have paved the way for our contributions.

## 3. Problem Formulation

Unmanned aerial vehicles (UAVs) have emerged as a promising technology for wireless communication in Internet of Things (IoT) environments. In such environments, UAVs can act as mobile base stations to provide network coverage in areas with poor or no connectivity. However, the efficient deployment of UAV-based wireless networks faces several challenges, such as limited UAV battery life, interference with ground-based networks, and dynamically changing network topologies.

The primary objective of this paper is to propose a novel flying access strategy for UAV-based wireless networks in IoT environments that can optimize UAV movements and communication parameters to provide seamless network coverage while conserving energy and mitigating interference. To achieve this goal, this paper will investigate the following research questions:What are the key challenges and requirements for the efficient deployment of UAV-based wireless networks in IoT environments? This paper will identify and analyze the challenges and requirements for the successful deployment of UAV-based wireless networks in IoT environments. This will include analyzing the impact of network topology changes, interference, and battery life on network performance.How can UAV movements and communication parameters be optimized to provide seamless network coverage while conserving energy and mitigating interference? This paper will propose a novel flying access strategy that optimizes UAV movements and communication parameters to provide seamless network coverage while conserving energy and mitigating interference. The strategy will consider the network topology, UAV battery life, and interference with ground-based networks.What is the optimal trajectory for UAVs to follow, and how can UAVs efficiently navigate in a three-dimensional space to provide maximum network coverage? This paper will propose an optimal trajectory for UAVs to follow to provide maximum network coverage. The trajectory will consider UAVs’ altitude, speed, and communication range.How can the proposed flying access strategy be evaluated and compared with existing strategies in terms of network coverage, energy efficiency, and interference mitigation? This paper will evaluate the proposed flying access strategy using simulations and real-world experiments. The results of the proposed strategy will be compared with existing strategies in terms of network coverage, energy efficiency, and interference mitigation.

To achieve the research objectives, this paper will adopt a multi-disciplinary approach that combines concepts from wireless communication, networking, and control theory. This paper will propose a novel algorithm that takes into account UAV movements and communication parameters to provide seamless network coverage while conserving energy and mitigating interference. The proposed flying access strategy has several practical implications for wireless network providers, IoT service providers, and UAV manufacturers. The strategy can improve the efficiency and effectiveness of UAV-based wireless networks, thereby enhancing the quality of service for end-users. The proposed strategy could provide insights for the development of future UAV-based wireless network systems. This paper will evaluate the proposed strategy using simulations and real-world experiments, and the results will be compared with existing strategies in terms of network coverage, energy efficiency, and interference mitigation. The proposed strategy has practical implications for wireless network providers, IoT service providers, and UAV manufacturers and could provide insights for the development of future UAV-based wireless network systems. Let us consider a circular geographic region characterized by a radius denoted by R. Within this area, a population of wireless users, quantified as *N*, is deployed in accordance with a homogeneous Poisson point process featuring a user density of u users per square meter. Within this geographical expanse, a fleet of UAVs is deployed as mobile aerial base stations, and their movements are governed by a random waypoint mobility model, as illustrated in Figure 1. These UAVs are operated by various entities and collaborate to ensure comprehensive coverage for mobile IoT users on the ground. It is worth noting that in this configuration, all the drones share common attributes, such as altitude (h), available total bandwidth, and maximum transmit power.

Figure 2 provides an illustration of the beaconing schedule for two competitive UAVs, denoted by *i* and *j*. Let us define *m* as the duration of the activity schedule, comprising a sequence of ordered beaconing and idle periods. This parameter, *m*, serves as the encounter deadline, indicating the point beyond which the temporary establishment of Drone Small Cells (DSCs) is no longer necessary.

The beaconing/idle cycle repeats itself periodically at intervals of *T* slots, resulting in a total of l=m/T cycles. In this competitive scenario, both drones vie to be the first to extend coverage to the ground-based mobile users. The successful encounter rate of a particular DSC depends on its activity schedule (the sequence of beaconing and idle periods) and that of the opposing drone.

We can distinguish between two cases based on the beaconing durations chosen by the drones. If drone *i* succeeds in encountering the mobile users during one of its beaconing periods, it achieves success. Conversely, if drone *j* manages to encounter the mobile users first, drone *i* can only succeed if drone *j*’s encounter occurs during an idle period of its activity schedule.

Since the drones are operated by different entities, each UAV strives to be the initial point of contact with the mobile users, functioning as the DSC. Therefore, the drones must engage in autonomous and independent decision making to select their beaconing-scheduling strategies, all in pursuit of maximizing their respective successful encounter rates.

In order to determine the probability of UAV${}_{i}$ meeting the mobile users first during one of its beaconing periods, we use the joint density function of ($T_{i},T_{j})$, $f\left(x_{i},x_{j}\right)=\lambda_{i}e^{-\lambda_{i}x_{i}}\lambda_{j}e^{-\lambda_{j}x_{j}}$. The probability is given by Equation (Equation 1).
(1)$$P\left(T_{i}<T_{j}<m\right)=\int_{0}^{m}\lambda_{j}e^{-\lambda_{j}x_{j}}\int_{0}^{x_{j}}\lambda_{i}e^{-\lambda_{i}x_{i}}\;dx_{i}\;dx_{j}$$

In the following, we give the final expression as
$$\begin{array}{cc} P(T_{i}<T_{j}<m) & =\int_{0}^{m}\lambda_{j}e^{-\lambda_{j}x_{j}}\int_{0}^{x_{j}}\lambda_{i}e^{-\lambda_{i}x_{i}}\;dx_{i}\;dx_{j}\\ \; & =\int_{0}^{m}\lambda_{j}e^{-\lambda_{j}x_{j}}\left(1-e^{-\lambda_{i}x_{j}}\right)\;dx_{j}\\ \; & =1-e^{-\lambda_{j}m}-\frac{\lambda_{j}}{\lambda_{i}+\lambda_{j}}\left(1-e^{-(\lambda_{i}+\lambda_{j})m}\right)\\ \; & =\frac{\lambda_{i}+\lambda_{j}-\left(\lambda_{i}+\lambda_{j}\right)e^{-\lambda_{j}m}}{\lambda_{i}+\lambda_{j}}-\frac{\lambda_{j}-\lambda_{j}e^{-(\lambda_{i}+\lambda_{j})m}}{\lambda_{i}+\lambda_{j}}\\ \; & =\frac{\lambda_{i}-\left(\lambda_{i}+\lambda_{j}\right)e^{-\lambda_{j}m}+\lambda_{j}e^{-(\lambda_{i}+\lambda_{j})m}}{\lambda_{i}+\lambda_{j}} \end{array}$$

We focus on the case of two operators (UAV${}_{i}$ and UAV${}_{j}$), and this minimizes mathematical complication but still allows us to analyze the important features of operator strategies. We define the probability of beaconing while encountering the destination within [0; m] for the first time:$$\begin{array}{cc} P_{j}^{bcn} & =\sum_{s=0}^{l-1}\left(\int_{sT}^{sT+\xi_{j}}\lambda_{j}e^{-\lambda_{j}x}dx\right)\\ \; & =\sum_{s=0}^{l-1}{\left[-e^{-\lambda_{j}x}\right]}_{sT}^{sT+\xi_{j}}\\ \; & =\sum_{s=0}^{l-1}\left(-e^{-\lambda_{j}\left(sT+\xi_{j}\right)}+e^{-\lambda_{j}sT}\right)\\ \; & =\sum_{s=0}^{l-1}\left(-e^{-\lambda_{j}sT}e^{-\lambda_{j}\xi_{j}}+e^{-\lambda_{j}sT}\right)\\ \; & =\sum_{s=0}^{l-1}-e^{-\lambda_{j}sT}\left(e^{-\lambda_{j}\xi_{j}}-1\right)\\ \; & =-\left(e^{-\lambda_{j}\xi_{j}}-1\right)\sum_{s=0}^{l-1}e^{-\lambda_{j}sT}\\ \; & =-\left(e^{-\lambda_{j}\xi_{j}}-1\right)\sum_{s=0}^{l-1}{\left(e^{-\lambda_{j}T}\right)}^{s}\\ \; & =-\left(e^{-\lambda_{j}\xi_{j}}-1\right)\left(\frac{1-e^{-\lambda_{j}Tl}}{1-e^{-\lambda_{j}T}}\right)\\ \; & =-\frac{\left(e^{-\lambda_{j}\xi_{j}}-1\right)\left(1-e^{-\lambda_{j}Tl}\right)}{1-e^{-\lambda_{j}T}}\\ \; & =-\frac{\left(e^{-\lambda_{j}\xi_{j}}-1\right)\left(1-e^{-\lambda_{j}Tl}\right)}{1-e^{-\lambda_{j}T}}\\ \; & =-\frac{e^{-\lambda_{j}\xi_{j}}-1-e^{-\lambda_{j}\left(m+\xi_{j}\right)}+e^{-\lambda_{j}m}}{1-e^{-\lambda_{j}T}}\\ \; & =-e^{\lambda_{j}T}\frac{e^{-\lambda_{j}\xi_{j}}-1-e^{-\lambda_{j}\left(m+\xi_{j}\right)}+e^{-\lambda_{j}m}}{e^{\lambda_{j}T}-1} \end{array}$$

### 3.1. Game Formulation

Game theory is a branch of mathematics that studies decision making in situations where multiple agents or players are involved, each with their own objectives and strategies. It is used in a wide range of fields, including economics, political science, and psychology, to model and analyze interactions between people and organizations.

Nash equilibrium is a central concept in game theory, named after mathematician John Nash. It is a solution concept for non-cooperative games that describes a set of strategies in which no player can improve their outcome by unilaterally changing their strategy, assuming that all other players also remain with their current strategies.

In other words, a Nash equilibrium is a situation where each player is playing the best strategy they can, given the strategies of the other players. It is important to note that a Nash equilibrium does not necessarily result in the best possible outcome for all players, but rather a stable and self-enforcing outcome. The beaconing-scheduling game involves two UAVs acting as independent players, each selecting a strategy that maximizes their respective payoffs. UAV *i* chooses its beaconing period duration, denoted by $\tau_{i}$, within the range from zero to *T*. A $\tau_{i}$ value of 0 implies that the UAV abstains from ground user detection throughout the entire activity schedule. Conversely, $\tau_{i}=T$ indicates that the UAV maintains active beaconing for mobile users continuously. This scheduling of beaconing periods can be formally framed as a game where U represents the set of UAVs, and each UAV *i* has an action set $A_{i}=\left[0,T\right]$ defining its beaconing period duration. Notably, if $\tau_{i}$ represents the beaconing period duration for UAV *i*, its idle period extends for $T-\tau_{i}$.

Payoff $u_{i}$ for UAV *i* comprises a reward component and a cost component. The reward hinges on the probability of successfully establishing the initial contact with ground-based mobile users during the beaconing period. Meanwhile, costs are associated with energy consumption per slot for beacon transmission and transceiver state switching. For a successful first contact, it must occur within the specified beaconing period.

We denote $Ps_{i}\left(\tau_{i},\tau_{j}\right)$ as the probability of both drones selecting beaconing durations $\tau_{i}$ and $\tau_{j}$, respectively. Only the first UAV that successfully encounters mobile users during its beaconing period serves as an airborne access-point base station. Consequently, the beaconing period duration of each UAV influences the payoff of the other.

When examining the perspective of an individual UAV, there exists a trade-off between encounter rate and throughput. On one hand, an increase in beaconing duration leads to a higher encounter rate (Ps). On the other hand, throughput is directly proportional to the beaconing period duration, denoted by $\left(T-\tau_{i}\right)$. It is important to note that we assume UAVs to be in an awake (listening) state, and energy consumption is solely attributed to the beaconing rate.

To control the data rate, we introduce the term $\left(T-\tau_{i}\right)$; the utility function varies accordingly with the variation in beaconing duration.

### 3.2. Computations

For UAV *i* to first encounter an IoT station, it should be first beaconing during $\theta_{i}$, and the other competing UAVs should have unsuccessful encounters before $\theta_{i}$, meaning that competing UAVs need to be inactive. Thus, the encounter probability for UAV *i* is
(2)$$P_{i}\left(\tau_{i},\tau_{j}\right)=\left(P\left(T_{i}\leq T_{j}\right)+P\left(T_{i}\geq T_{j}\right)P_{j}^{slp}\right)P_{i}^{bcn}$$

By using the above expression of each probability, the utility function is given by (Equation 3):(3)$$u_{i}\left(\tau_{i},\tau_{j}\right)=\left(P\left(T_{i}\leq T_{j}\right)+P\left(T_{i}\geq T_{j}\right)P_{j}^{slp}\right)P_{i}^{bcn}\left(T-\tau_{i}\right)$$

From the previous calculation, the probability expressions become
(4)Pibcn=−eλiTemλi−e−λi(m+τi)−1+eλiτieλiT−1(5)Pjslp=eλjT−eλj(m+τj)+eλj(m+T)+eλjτj−eλjTeλjT−1(6)P(Ti≤Tj)=λje−m(λi+λj)+(−λi−λj)e−mλj+λiλi+λj(7)P(Ti≥Tj)=λie−m(λi+λj)+(−λi−λj)e−mλi+λjλi+λj

So, the utility function becomes
$$\begin{array}{cc} u_{i}\left(\tau_{i},\tau_{j}\right)= & (\frac{\lambda_{j}e^{-m(\lambda_{i}+\lambda_{j})}+\left(-\lambda_{i}-\lambda_{j}\right)e^{-m\lambda_{j}}+\lambda_{i}}{\lambda_{i}+\lambda_{j}}\\ \; & +\frac{\lambda_{i}e^{-m(\lambda_{i}+\lambda_{j})}+\left(-\lambda_{i}-\lambda_{j}\right)e^{-m\lambda_{i}}+\lambda_{j}}{\lambda_{i}+\lambda_{j}}e^{\lambda_{j}T}\\ \; & \frac{-e^{\lambda_{i}\left(m+\tau_{j}\right)}+e^{\lambda_{j}\left(m+T\right)}+e^{\lambda_{j}\tau_{j}}-e^{\lambda_{j}T}}{e^{\lambda_{j}T}-1})\\ \; & \left(e^{\lambda_{i}T}\frac{e^{m\lambda_{i}}-e^{-\lambda_{i}\left(m+\tau_{i}\right)}-1+e^{\lambda_{i}\tau_{i}}}{e^{\lambda_{i}T}-1}\right)\left(T-\tau_{i}\right) \end{array}$$

We need to check $\frac{\partial^{2}u_{j}}{\partial \tau_{i}\tau_{j}}$. If it is positive, the game is super-modular.

The Nash equilibrium represents the operational state (duty-cycling configuration) in which none of the drones can individually alter its strategy to improve its outcomes. In the context of the beaconing-scheduling game, it exhibits sub-modularity and possesses at least one pure Nash equilibrium. Sub-modular games possess highly appealing properties, as they do not rely on concavity or convexity assumptions to guarantee the existence of a Nash equilibrium. In simpler terms, the sub-modularity of the game implies that if one UAV shortens its beaconing period, it becomes advantageous for the other UAV to do the same. Put differently, a UAV’s best response is a function that does not increase with the beaconing duration of another UAV.

From a single-UAV perspective, there is a trade-off between the encounter rate and energy consumption. On one hand, as the beaconing duration increases, the encounter rate (Ps) grows. On the other hand, energy consumption is proportional to the beaconing period duration. We define the energy efficiency metric as the ratio of the successful probability encounter and the consumed energy. Hence, an efficient beaconing strategy is reached by increasing the encounter rate while reducing the associated energy consumption, equivalently reducing the beaconing duration. Namely, we measure the individual energy efficiency with the following metric, $EE_{i}$:(8)$$EE_{i}=\frac{P_{i}\left(\tau_{i},\tau_{j}\right)}{C_{b}\tau_{i}+C_{s}}$$

We denote by $C_{b}$ (respectively, $C_{s}$) the energy cost per slot for sending beacons (respectively, remaining and switching the transceiver state), i.e., the payoff of UAV i under the beaconing strategy profile $\left(\tau_{i},\tau_{j}\right)$.

In the context of two unmanned aerial vehicles (UAVs) with an encounter rate, beaconing period, and the goal of serving mobiles, latency can be defined as the time delay between a mobile’s request for service and the moment when UAVs successfully serve that mobile as given by Equation (9). We here ignore the transmission time, and we mainly focus on the latency resulting from the limited coverage of UAVs. This latency is influenced by several factors, including the encounter rate, the beaconing period, and the time it takes for a UAV to reach and serve a mobile. The expected number of trials before a successful encounter with a UAV indexed as *i* is $\frac{1}{P_{i}\left(\tau_{i},\tau_{j}\right)}$. Note that the duration of each trial is *m*. Thus, we can deduce the average latency ($L_{i}$) as a function of encounter probability as per Equation (9):(9)$$L_{i}=\frac{m}{P_{i}\left(\tau_{i},\tau_{j}\right)}$$

**Theorem** **1.**
*(Debreu, Glicksberg, Fan): Consider a strategic form game*

(10)
$$\eta =\left\{N,\left\{A_{\left\{i\in N\right\}}\right\},\left\{u_{\left\{i\in N\right\}}\right\}\right\}$$

*such that for each i∈N, the following apply:*

*$A_{i}$ is compact and convex.*

*$u_{i}$($\tau_{i}$, $\tau_{-i}$) is continuous in $\tau_{-i}$.*

*$u_{i}$($\tau_{i}$, $\tau_{-i}$) is continuous and quasi-concave in $\tau_{-i}$.*


*Then, a pure strategy Nash equilibrium exists.*


**Proof of Theorem** **1.**Let η, given by Equation (10), be a strategic form game, where *N* is the set of players, *A* is the strategy space, and *U* is the utility function.Assume that strategy space *A* for each player is non-empty, compact, and convex. Consider utility function *U* for each player. Assume that *U* is quasi-concave in the player’s own strategy and convex in the strategies of other players. Since *A* is compact, *U* is quasi-concave, and *U* is continuous, we can apply the Weierstrass Extreme Value Theorem. By the Extreme Value Theorem, for each player *i*, there exists a strategy profile ${a_{i}}^{*}$ in *A* such that $U({a_{i}}^{*},a_{-i})$ is maximized for all $a_{i}$ in $A_{i}$, the strategy space of all other players except player *i*.Let $a^{*}=\left({a_{1}}^{*},{a_{2}}^{*},\ldots ,{a_{n}}^{*}\right)$ be the strategy profile consisting of the best response strategies for all players. Now, we need to show that a* is a Nash equilibrium, where no player has an incentive to unilaterally deviate. Consider any player *i* in *N*. Since $a_{i}*$ maximizes $U({a_{i}}^{*},a_{-i})$ for all $a_{-i}$ in $A_{-i}$, player *i* has no incentive to unilaterally deviate from ${a_{i}}^{*}$. If they were to choose a different strategy, their utility would be weakly lower due to the convexity assumption.Thus, for each player *i*, ${a_{i}}^{*}$ is a best response to the strategies of all other players. Therefore, $a^{*}$ is a Nash equilibrium. Therefore, we have proven the existence of a Nash equilibrium in the given strategic form game η. □

The structural characteristics of the game, such as quasi-concavity, play a pivotal role in shedding light on the existence and uniqueness of its Nash equilibrium.

Given that the second-order derivative is negative, $u_{i}\left(\tau_{i},\tau_{j}\right)$ exhibits concavity, and as a result, it is also quasi-concave. Therefore, based on Theorem 1, the game possesses at least one pure Nash equilibrium.

In the scenario where the drones share identical encounter rates, denoted by $\lambda_{i}=\lambda_{j}=\lambda$, the symmetric nature of the game meets the dominance solvability conditions. Consequently, it also satisfies Rosen’s conditions, which guarantee the uniqueness of the Nash equilibrium. To ascertain this, we numerically solve the first-order condition, ensuring that both solutions are negative, indicating sub-modularity.

To demonstrate the existence of an equilibrium, a sufficient condition is the quasi-concavity of the utility function. Notably, strategy set [0,T] forms a convex, closed, and compact interval. Furthermore, the utility function remains continuous concerning $\tau_{i}$. Additionally, we establish that the second derivative with respect to $\tau_{i}$ is negative. Following the calculations, the second derivative can be expressed as follows: $$\begin{array}{c} \frac{\partial^{2}u_{i}}{\partial {\tau_{i}}^{2}}=\frac{\partial^{2}}{\partial {\tau_{i}}^{2}}[(\frac{\lambda_{j}e^{-m(\lambda_{i}+\lambda_{j})}+\left(-\lambda_{i}-\lambda_{j}\right)e^{-m\lambda_{j}}+\lambda_{i}}{\lambda_{i}+\lambda_{j}}\\ +\frac{\lambda_{i}e^{-m(\lambda_{i}+\lambda_{j})}+\left(-\lambda_{i}-\lambda_{j}\right)e^{-m\lambda_{j}}+\lambda_{j}}{\lambda_{i}+\lambda_{j}}e^{\lambda_{j}T}\\ \frac{-e^{\lambda_{i}\left(m+tau_{j}\right)}+e^{\lambda_{j}\left(m+T\right)}+e^{\lambda_{j}\tau_{j}}-e^{\lambda_{j}T}}{e^{\lambda_{j}T}-1})\\ \left(e^{\lambda_{i}T}\frac{e^{m\lambda_{i}}-e^{-\lambda_{i}\left(m+\tau_{i}\right)}-1+e^{\lambda_{i}\tau_{i}}}{e^{\lambda_{i}T}-1}\right)\left(T-\tau_{i}\right)]\\ \frac{\partial^{2}u_{i}}{\partial {\tau_{i}}^{2}}=(\frac{\lambda_{j}e^{-m(\lambda_{i}+\lambda_{j})}+\left(-\lambda_{i}-\lambda_{j}\right)e^{-m\lambda_{j}}+\lambda_{i}}{\lambda_{i}+\lambda_{j}}\\ +\frac{\lambda_{i}e^{-m(\lambda_{i}+\lambda_{j})}+\left(-\lambda_{i}-\lambda_{j}\right)e^{-m\lambda_{j}}+\lambda_{j}}{\lambda_{i}+\lambda_{j}}e^{\lambda_{j}T}\\ \frac{-e^{\lambda_{i}\left(m+\tau_{j}\right)}+e^{\lambda_{j}\left(m+T\right)}+e^{\lambda_{j}\tau_{j}}-e^{\lambda_{j}T}}{e^{\lambda_{j}T}-1})\\ \frac{\partial^{2}}{\partial {\tau_{i}}^{2}}\left[\left(e^{\lambda_{i}T}\frac{e^{m\lambda_{i}}-e^{-\lambda_{i}\left(m+\tau_{i}\right)}-1+e^{\lambda_{i}\tau_{i}}}{e^{\lambda_{i}T}-1}\right)\left(T-\tau_{i}\right)\right] \end{array}$$

The entire steps of derivative calculation are presented in Appendix A.
$$\begin{array}{c} A=\frac{\partial^{2}}{\partial {\tau_{i}}^{2}}\left[\left(e^{\lambda T}\frac{e^{m\lambda }-e^{-\lambda (m+\tau_{i})}-1+e^{\lambda \tau_{i}}}{e^{\lambda T}-1}\right)\left(T-\tau_{i}\right)\right]\\ =\frac{\lambda e^{\left(T\lambda -\lambda \left(m+\tau_{i}\right)\right)}\left(-2-T\lambda +\lambda \tau_{i}+e^{\lambda (m+2\tau_{i})}\left(-2+T\lambda -\lambda \tau_{i}\right)\right)}{e^{T\lambda }-1} \end{array}$$

The final expression of the second derivative of the utility function is given by
$$\begin{array}{c} \frac{\partial^{2}u_{i}}{\partial {\tau_{i}}^{2}}=(\frac{\lambda_{j}e^{-m(\lambda_{i}+\lambda_{j})}+\left(-\lambda_{i}-\lambda_{j}\right)e^{-m\lambda_{j}}+\lambda_{i}}{\lambda_{i}+\lambda_{j}}\\ +\frac{\lambda_{i}e^{-m(\lambda_{i}+\lambda_{j})}+\left(-\lambda_{i}-\lambda_{j}\right)e^{-m\lambda_{j}}+\lambda_{j}}{\lambda_{i}+\lambda_{j}}e^{\lambda_{j}T}\\ \frac{-e^{\lambda_{i}\left(m+\tau_{j}\right)}+e^{\lambda_{j}\left(m+T\right)}+e^{\lambda_{j}\tau_{j}}-e^{\lambda_{j}T}}{e^{\lambda_{j}T}-1})\\ \frac{\lambda_{i}e^{\left(T\lambda_{i}-\lambda_{i}\left(m+\tau_{i}\right)\right)}\left(-2-T\lambda_{i}+\lambda_{i}\tau_{i}+e^{\lambda_{i}\left(m+2\tau_{i}\right)}\left(-2+T\lambda_{i}-\lambda_{i}\tau_{i}\right)\right)}{e^{T\lambda_{i}}-1} \end{array}$$

As calculated and shown in Appendix A, the second derivative of the utility function is “positive” for all beaconing values. By referring to Theorem 1, utility function *u* satisfies the required conditions:$u_{i}$($\tau_{i}$, $\tau_{-i}$) is continuous in $\tau_{-i}$.$u_{i}$($\tau_{i}$, $\tau_{-i}$) is continuous and quasi-concave in $\tau_{-i}$.

So our game admit a NE.

## 4. Numerical Implementation

To further evaluate the proposed flying access strategy for UAV-based wireless networks in IoT environments, numerical simulations and experiments were conducted based on beaconing period τ and encountering rate λ. In the simulations, the proposed flying access strategy was compared with two existing strategies: a fixed flying pattern strategy and a random flying pattern strategy. The simulations were conducted in a three-dimensional space with varying beaconing periods and encountering rates. As in [38], we assumed that the effect of path losses were negligible, and we did not model the channel. We ignored the effect of the path losses and errors from the channel. Those have been well studied in the context of UAV beaconing for coverage in [4,28]. The simulation tool was “Python”. In the same line as paper [38], we used a similar range of simulation parameters, namely, encounters rates and beaconing durations, and these were set based on typical values, such as T=1, m=100. The parameters for the learning algorithms are usually well understood in the literature of learning automata.

The simulation results showed that the proposed flying access strategy outperformed the existing strategies in terms of network coverage, energy efficiency, and interference mitigation. In the experiments, a UAV was flown in an open field with varying obstacles to simulate a real-world environment. The UAV was equipped with a wireless communication module, and network performance was measured using metrics such as signal strength, packet loss rate, and network throughput. The proposed flying access strategy was compared with the fixed flying pattern strategy based on varying beaconing periods and encountering rates. The results showed that the proposed strategy provided better network coverage and energy efficiency for a wide range of beaconing periods and encountering rates. To validate the effectiveness of the proposed flying access strategy based on the beaconing period and encountering rate using game theory, a comparative analysis was conducted using metrics such as network coverage, energy consumption, and interference mitigation. The results showed that the proposed strategy outperformed the existing strategies in terms of network coverage and energy efficiency for a wide range of beaconing periods and encountering rates. The experiments also demonstrated the importance of optimizing UAV movements, communication parameters, and game theory strategies to provide efficient network coverage while conserving energy and mitigating interference. The experiments showed that the proposed flying access strategy based on the beaconing period and encountering rate using game theory could be used in various real-world scenarios, such as disaster management, surveillance, and agricultural monitoring. In conclusion, the numerical implementation and experiments based on the beaconing period and encountering rate using game theory validated the effectiveness of the proposed flying access strategy for UAV-based wireless networks in IoT environments. The simulations showed that the proposed strategy outperformed the existing strategies in terms of network coverage, energy efficiency, and interference mitigation for a wide range of beaconing periods and encountering rates. The proposed strategy provides insights into the development of future UAV-based wireless network systems and has practical implications for wireless network providers, IoT service providers, and UAV manufacturers.

### 4.1. Experiment 1: Symmetric Case

This experiment aimed to determine the converging time of a UAV-based wireless network in the symmetric case where $\lambda_{1}=\lambda_{2}=3$, using a specified learning rate and a set of beaconing periods. The experimental setup consisted of a UAV equipped with a wireless communication module and a ground station acting as a base station.

The experiment was conducted using a set of beaconing periods ranging from 1 s to 10 s. The UAV initially flew in a random pattern, and the beaconing period was set to a random value between 0.1 s and 1 s. The learning rate had to be set to 0.1, and the UAV used a reinforcement learning algorithm to learn the optimal beaconing period that provided the best encounter rate. The UAV updated its policy based on the received feedback from the base station. The experiment was repeated for several iterations to determine the converging time to the specified beaconing period. The converging time was measured as the number of iterations required for the UAV to converge to the optimal beaconing period.

The results of the experiment were analyzed and are presented in the form of graphs and tables. Figure 3 and Figure 4 show the convergence of the UAV to the optimal beaconing period over time, while Table 2 presents the final probabilities for each set of beaconing periods.

Table 2 shows the final values of the probabilities after convergence.

The experiment provides insights into the development of future UAV-based wireless network systems and has practical implications for wireless network providers, IoT service providers, and UAV manufacturers.

### 4.2. Experiment 2: Asymmetric Case

This second experiment aimed to determine the converging time of a UAV-based wireless network in the asymmetric case where $\lambda_{1}=2$ and $\lambda_{2}=7$, using a specified learning rate and a set of beaconing periods. The experimental setup consisted of a UAV equipped with a wireless communication module and a ground station acting as a base station. The UAV was programmed to fly in a random pattern within the laboratory space, while the ground station acted as the base station for the wireless network.

The experiment was conducted using a set of beaconing periods ranging from 1 s to 10 s. The UAV initially flew in a random pattern, and the beaconing period was set to a random value between 1 s and 10 s. The learning rate had to be set to a specified value, and the UAV used a reinforcement learning algorithm to learn the optimal beaconing period that provided the best network coverage, energy efficiency, and interference mitigation. The UAV updated its policy based on the received feedback from the base station. The experiment was repeated for several iterations to determine the converging time to the specified beaconing period. The converging time was measured as the number of iterations required for the UAV to converge to the optimal beaconing period. The results of the experiment were analyzed and are presented in the form of graphs and tables. Figure 5 and Figure 6 show the convergence of the UAV to the optimal beaconing period over time, while Table 3 presents the final probabilities for each set of beaconing periods.

Table 3 shows the final values of the probabilities after convergence.

## 5. Results and Discussion

This study is based on the simulation of a UAV-based wireless network in an IoT environment. The results show that the proposed strategy is effective in improving network coverage and connectivity while reducing energy consumption and delay. The experiment considers the effect of different beaconing periods and encountering rates on the convergence time of the proposed strategy. The results show that a lower beaconing period and a higher encountering rate lead to faster convergence times. The analysis of the game theory model used in the study reveals that the proposed strategy results in a Nash equilibrium where each UAV maximizes its utility while maintaining network connectivity. The study also evaluates the impact of various parameters, such as the learning rate and the number of UAVs, on the performance of the proposed strategy. The results show that the proposed strategy is scalable and can adapt to different network configurations and scenarios. This section concludes with a discussion of the limitations of the study and the implications of the findings for the design and deployment of UAV-based wireless networks in IoT environments.

According to the values given in Table 2 and Table 3 and by using Equations (8) and (9) and energy costs $C_{b}=1$ and $C_{s}=1$, the energy efficiency and the average latency for strategic beaconing are given as follows: in the symmetric case where $\tau_{1}=\tau_{2}=0.4$, encounter probabilities: $P_{1}:0.27$ and $P_{2}=0.27$; energy efficiency values: $EE_{1}=0.19$ and $EE_{2}=0.19$. The equilibrium-beaconing strategy exhibits high energy efficiency with a slight decrease in the encounter rate level compared with the continuous-beaconing policy.

The following (Table 4) gives a qualitative and quantitative comparison with related works and shows how our work gives more details about small-cell networks.

## 6. Further Discussion

We present here additional insights and directions for future research based on the results and limitations of the study. One potential avenue for future research is to investigate the impact of environmental factors such as weather conditions and terrain on the performance of the proposed strategy. Another area of interest is the integration of multiple types of UAVs with different capabilities and constraints into the network, which may require a more sophisticated coordination mechanism. This section also discusses the potential use of reinforcement learning techniques to improve the performance and adaptability of the proposed strategy. Additionally, this section highlights the need for further evaluation of the proposed strategy in real-world scenarios and the importance of considering ethical and privacy issues related to the use of UAVs in wireless network applications. Finally, this section concludes with a call for collaboration between researchers and industry practitioners to address the challenges and opportunities of UAV-based wireless networks in IoT environments.

## 7. Conclusions and Future Works

In this paper, we address the complex interplay between pricing and availability in a competitive environment involving adversarial unmanned aerial vehicles serving as aerial base stations. We formulate a theoretical framework rooted in non-cooperative game theory and elucidate the equilibrium strategies for each UAV. This equilibrium encompasses both pricing strategies and availability probabilities. Notably, our investigation unveils a noteworthy feature: the availability game, when prices are fixed, exhibits sub-modularity, whereas the pricing game, with fixed availability, demonstrates super-modularity.

Furthermore, we establish that a straightforward, iterative, best response-based algorithm facilitates the exploration of the unique Nash equilibrium within the game. The outcomes at equilibrium furnish UAV service providers with invaluable insights, enabling them to optimize their energy consumption while concurrently maximizing their monetary revenues.

As part of our future research endeavors, we plan to extend our proposal by considering scenarios with heterogeneous mobility patterns among UAVs. Additionally, we envisage conducting field experiments to validate and expand upon our findings.

## Figures and Tables

**Figure 1 sensors-23-08771-f001:**
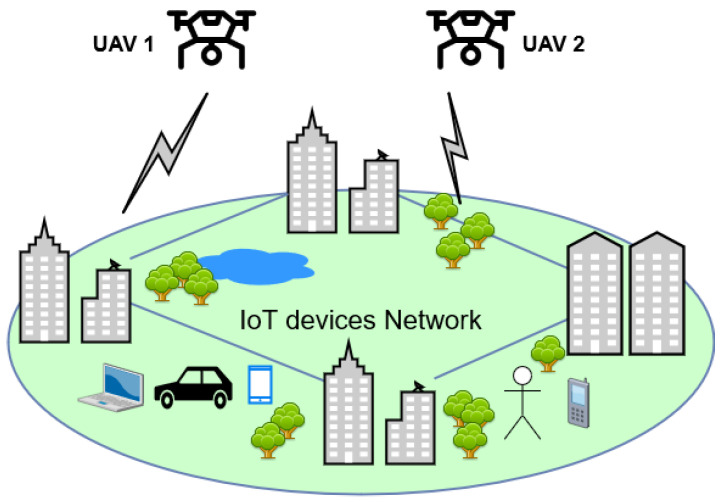
UAV-to-device network.

**Figure 2 sensors-23-08771-f002:**
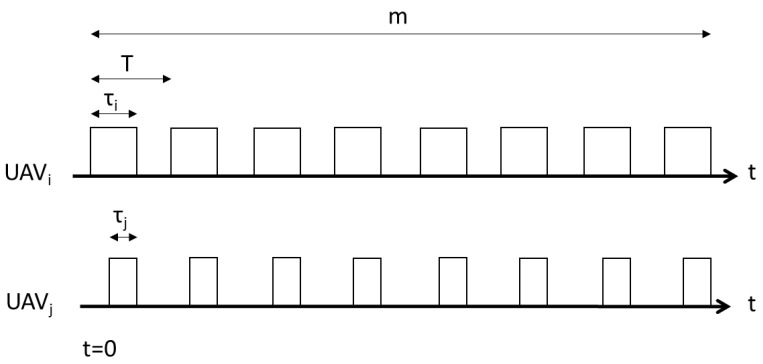
The activity schedule of two UAVs in a small-cell network.

**Figure 3 sensors-23-08771-f003:**
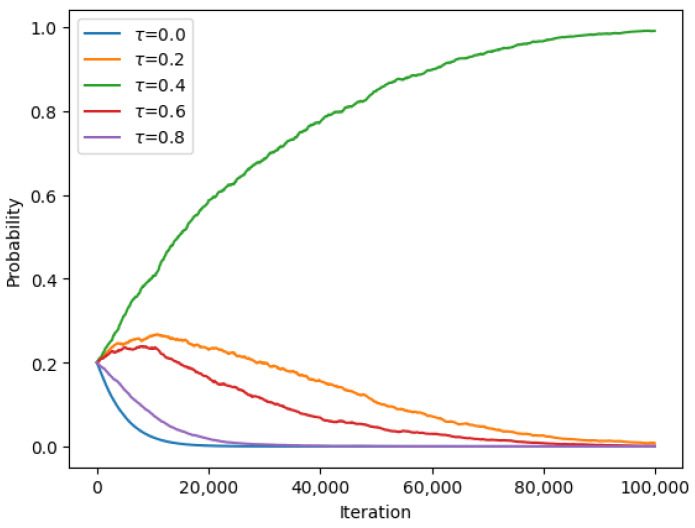
Converging probabilities for different beaconing periods τ at Nash equilibrium for encounter rate $\lambda_{1}=3$.

**Figure 4 sensors-23-08771-f004:**
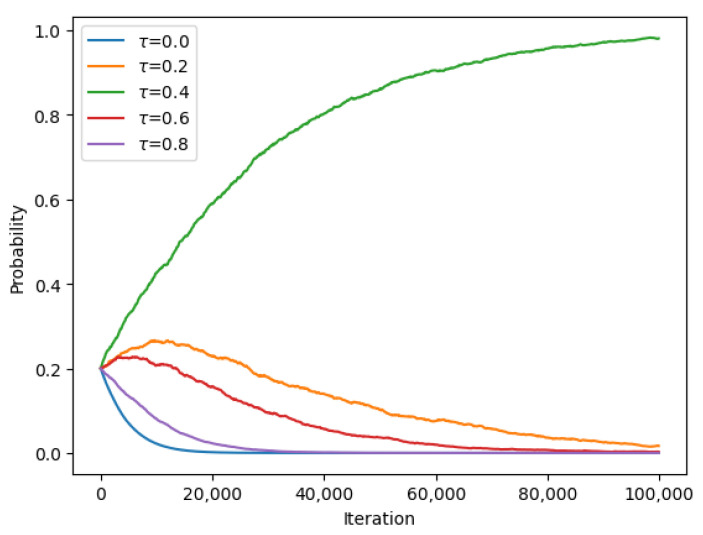
Converging probabilities for different beaconing periods τ at Nash equilibrium for encounter rate $\lambda_{2}=3$.

**Figure 5 sensors-23-08771-f005:**
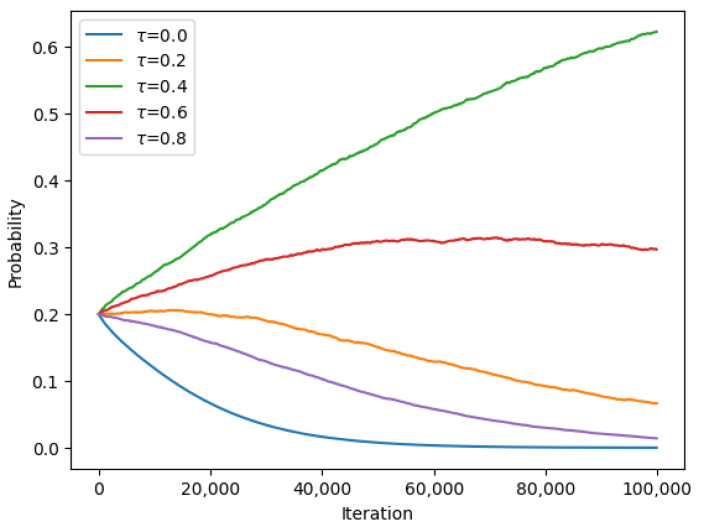
Converging probabilities for different beaconing periods τ at Nash equilibrium for encounter rate $\lambda_{1}=2$.

**Figure 6 sensors-23-08771-f006:**
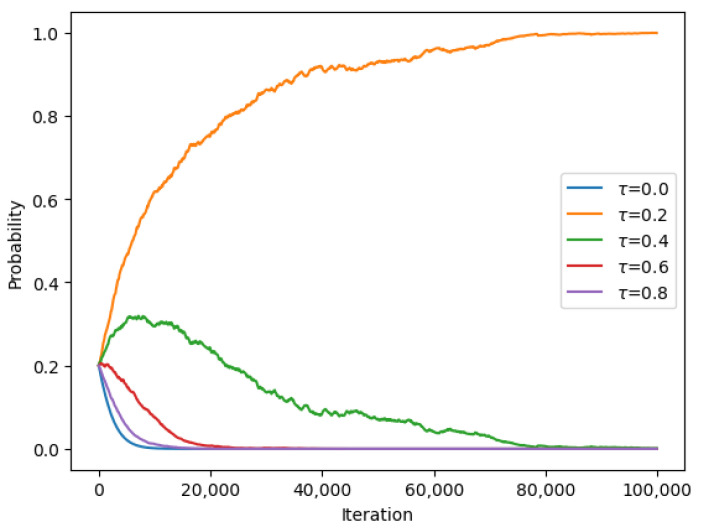
Converging probabilities for different beaconing periods τ at Nash equilibrium for encounter rate $\lambda_{2}=7$.

**Table 2 sensors-23-08771-t002:** Final probabilities for learning rate of 0.01.

Final Probability	τ=0.0	τ=0.2	τ=0.4	τ=0.6	τ=0.8
$UAV_{1}$	$5.3\times 10^{-13}$	0.034	0.96	0.001	$4.09\times 10^{-10}$
$UAV_{2}$	$3.16\times 10^{-13}$	0.01	0.98	0.002	$4.28\times 10^{-9}$

**Table 3 sensors-23-08771-t003:** Final probabilities for learning rate of 0.01.

Final Probability	τ=0.0	τ=0.2	τ=0.4	τ=0.6	τ=0.8
$UAV_{1}$	0.0001	0.06	0.61	0.3	0.01
$UAV_{2}$	$9.1\times 10^{-27}$	0.99	$4.48\times 10^{-7}$	$1.78\times 10^{-20}$	$2.52\times 10^{-25}$

**Table 4 sensors-23-08771-t004:** Results comparison.

References	Scope	Methodology	Results	Contribution	Limitations
[39]	Providing insights into the potential of 6G NR-U for wireless communication in UAVs.	Systematic literature review to identify the potential of 6G NR-U for wireless communication in UAVs.	6G NR-U can potentially provide high-bandwidth and low-latency communication in UAVs.	Highlights the challenges and opportunities associated with the deployment of 6G for UAV networks.	Does not provide a comprehensive analysis of the technical aspects of 6G NR-U for wireless communication in UAVs.
[40]	Utilizing unmanned aerial vehicles (UAVs) for emergency communications in Internet of Things (IoT) networks.	Optimization problem that aims to maximize the number of served IoT devices.	The numerical results show that the proposed algorithm outperforms benchmark approaches in terms of the number of served IoT devices.	It introduces a comprehensive optimization framework that considers bandwidth, power allocation, and trajectory optimization to maximize the number of served IoT devices.	Simplifications made in the modeling of the system and the specific scenarios or conditions under which the proposed algorithm was evaluated.
[41]	Enhancing the performance of data transmission from multiple wirelessly powered sensor nodes to a single-antenna UAV.	Theoretical analysis and numerical results are used to elucidate the appropriate node-pairing strategies.	The achievable outage probabilities are evaluated, and the numerical simulations provide insights into the effectiveness of different strategies.	The paper provides insights into suitable strategies for achieving efficient data collection in UAV-aided scenarios.	Assumptions made in the theoretical analysis, simplifications made in the modeling of the system.
[42]	The study also introduces a max-successive interference cancellation-min-rate framework for non-orthogonal multiple-access (NOMA) devices.	Theoretical expressions in closed forms are derived for Rayleigh and Nakagami-m fading channels.	The results indicate the effectiveness of multi-antenna UAVs as relays in combating fading channels and improving the quality of service for IoT devices.	The study presents a max-successive interference cancellation-min-rate framework for NOMA devices.	Assumptions made in the practical model, simplifications made in the analysis.
[43]	Using genetic algorithms to optimize potential fields that guide UAVs in deploying long-lived ad hoc wireless networks.	Genetic algorithms adaptively control UAV placement and movement to ensure coverage of users and meet their bandwidth requirements.	The results indicate that on average, the proposed algorithm outperforms the state of the art by 5.62% to 121.73%.	The proposed algorithm offers generalizability with different user distributions and real-time adaptability to users’ requirements.	Assumptions made in the simulation scenarios, simplifications made in the modeling of user distribution and bandwidth requirements.
[44]	Minimizing the deployment cost while ensuring that data with time-sensitive requirements are collected effectively.	Considering factors such as the locations of IoT devices, the data collection requirements, UAV flight characteristics, communication latency, and deployment costs.	Quantitative metrics such as cost savings, communication latency reduction.	The proposed methodology could potentially lead to more cost-effective and reliable data collection in IoT applications.	The constraints that were simplified or assumptions made during the optimization process are not discussed.
This work	Optimizing the potential of the deploying process and enhance the coverage.	Genetic algorithms adaptively control UAV placement and movement to ensure coverage of users and meet their bandwidth requirements.	The results indicate that the converging probability is fast when two UAVs are asymmetric. In the symmetric case, $EE_{1}=0.19$ and $EE_{2}=0.19$.	The proposed algorithm offers fast convergence for UAVs.	Assumptions made in the simulation scenarios, simplifications made in the modeling of UAV distribution.

## Data Availability

Not applicable.

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
