# Peer review of "Coverage Strategy for Small-Cell UAV-Based Networks in IoT Environment"

_sensors, 2023, doi:10.3390/s23218771_

Round 1

Reviewer 1 Report

The article needs some work:

In the Results and Discussion section the authors claim that they have carried  out a large amount of numerical calculations to study energy efficiency and data transmission efficiency. However, in this section they provide qualitative comparisons only and do not provide quantitative estimations. As a result, I, as a reader, do not have a comprehensive picture of the advantages of the proposed strategy. I would recommend that the authors rework this section to include quantitative estimations.

1. For example, it will be of interest to the reader if the authors provided a general analysis in this section, presenting quantitative estimations of how effective the strategy is and by what numbers power consumption and latency are reduced.

2. What is the optimal number of UAVs and what is the learning rate in that case?

3. In reality, scaling and adaptation cannot be done indefinitely, so limitations and underlying reasons for those limitations should be specified.

4. In Table 4, in which the authors compare their results with the work of other researchers, it is also necessary to provide quantitative estimations.

5. The authors claim that “the results indicate that, on average, the proposed algorithm outperforms the state of the art by 5.62% to121.73%.” This result requires detailed discussion, such as in which case does the proposed algorithm outperform the state-of-the-art by 121.73% and why?

Reviewer 2 Report

The authors focus on an efficient energy consumption optimization scheduling. They provide a sub-modular game perspective of the problem and investigates its structural properties, while provide a learning algorithm that ensures convergence of the considered network to a Nash equilibrium operating point. This work is meaningful for IoT network. There are some comments below which I recommend to give one chance to take a revision. A more comprehensive literature survey may be provided with multi visual GRU based survivable computing power scheduling in metro optical networks, brainIoT: brain-like productive services provisioning with federated learning in industrial IoT, socially-aware traffic scheduling for edge-assisted metaverse by deep reinforcement learning. 

Reviewer 3 Report

Although the paper deals with a vary interesting topic, some issues need to be clarified prior publication:

1) There are very few details provided on the overall simulation setup. Which simulation tool have you used? What are the main simulation parameters and assumption? How are pathlosses and channel modelled?

2) In the same context, justification of parameters is essential.
